# Influence of Cold Deformation on Carbide Precipitation Kinetics in a Fe-22Mn-0.45C TWIP Steel

**DOI:** 10.3390/ma15113748

**Published:** 2022-05-24

**Authors:** Javier Escobar, José Luis Jiménez, Alfredo Artigas, Juan Perez-Ipiña, Alberto Monsalve

**Affiliations:** 1Department of Metallurgical Engineering, Universidad de Santiago de Chile, Av. Ecuador 3735, Estación Central, Santiago 9170124, Chile; escobar.javier91@gmail.com (J.E.); jimenezpinojoseluis@gmail.com (J.L.J.); alfredo.artigas@usach.cl (A.A.); 2Conicet, Neuquén 8300, Argentina; pipinajuan@gmail.com; 3Structural Integrity Program, Engineering Faculty, Universidad de Santiago de Chile, Santiago 9170124, Chile

**Keywords:** TWIP steels, plastic deformation, Fe-Mn-carbides, precipitation kinetics

## Abstract

The carbide precipitation kinetics in a Fe-22Mn-0.45C TWIP steel subjected to three different cold-deformation levels, annealed at various temperatures, were studied. The studied carbides included chemical compositions, morphology, precipitation sites, volume fraction, and size. Manganese carbides were precipitated in a temperature range between 525 and 650 °C. Volume fraction increased with cold-deformation and decreased with annealing temperature. Carbide size increased with cold-deformation and annealing temperatures up to 625 °C, suffering a notable reduction at 650 °C. Precipitation kinetics were described by means of precipitation curves for 0.1% (vol.) of Fe-Mn-carbides. A kinetic model was used, and two stages were found. Complementarily, austenite grain size and microhardness were also measured. With increases in annealing time, microhardness decreased until it reached a nearly constant value, indicating that recrystallization was complete, while, with increases in annealing temperature, grain size increased.

## 1. Introduction

In recent years, TWinning Induced Plasticity (TWIP) high-manganese steels have been developed in the context of Advanced High-Strength Steels (AHSS), as they have an extraordinary combination of high yield and ultimate stresses with high ductility. These fully austenitic steels reach UTS values between 800 and 1100 MPa, with up to 95% total elongation to fracture at room temperature, mainly depending on chemical composition, stacking fault energy and thermo-mechanical treatment [1]. These properties, obtained by the contribution of twinning and dislocation slip interactions, can be translated into exceptional formability, producing lightweight and strong structural parts.

In the context of high-Mn steels, low-carbon high-Mn steels are also important in material science. In this kind of steel, the precipitation of carbides also occurs, although the number of carbides is lower than in the case of high C-high Mn steels. The presence of carbon in steels increases their strength; however, in high-Mn steels, such as the steel studied in this work, C increases the stacking fault energy (SFE), which influences the mechanical behavior of steel. Specifically, when SFE is between 15 and 45 mJ/m^2^ [1], twinning occurs during plastic deformation. Regarding the alloying elements, Nb, Ti and V, among others, are frequently added to some steels as strong carbide formers [2,3,4].

The automotive industry has adapted, by using new types of steel with better mechanical properties, to the higher standards caused by the ever-growing need to enhance passengers’ safety and decrease emissions by reducing weight to attain a lower fuel consumption. TWIP steels have a high capacity to absorb significant amounts of energy on deformation, typically double than that of other high-strength steels [5]. Twinning increases the work-hardening rate by acting as an obstacle to gliding dislocations [6].

Several aspects of TWIP steels have been studied, such as the basic phenomena and possible models, relationship between work-hardening and twinning [6], the effect of texture in work-hardening [7], effect of microstructure in the formation of twins [8], hardening mechanisms [9,10], mechanical behavior in both cold and hot conditions [11,12], the twinning phenomena and its dependence on temperature, the effect of different alloying elements [13], the microstructure and texture components that appear under different conditions [14], the recrystallization and grain growth that determine kinetics laws, the mechanical response in different tests such as hot tensile tests [15], high strain rate [16] and the fatigue response [17].

Nevertheless, several aspects that strongly affect their mechanical behavior are not completely known to date; for example, when subjected to different temperature conditions, the precipitation of Fe-Mn-carbides may adversely affect their mechanical properties [18]. Bouaziz et al. (2011) [19] and Scott et al. (2006) [20], stated that the occasional presence of carbides in TWIP steels reduces the C concentration in solid solution, thus lowering the SFE, and that the cementite precipitated at grain boundaries also has a strong and detrimental effect on the toughness and ductility. Kang et al. [21] studied the effect of recrystallization annealing temperature on the precipitation behavior of manganese carbides in Fe-18Mn-0.6C-1.5Al TWIP steel, and found that the precipitation kinetics showed a typical C-curve with a nose temperature at 800 °C. Scott et al. (2011) [22] studied the precipitation of Nb, V and Ti carbides in high-Mn TWIP steels and their effect on the steel strengthening, finding that, when deformation < 25%, precipitates have no effect on work-hardening. Moreover, Cancino Serrano showed that they can precipitate during thermomechanical forming or welding processes [23] when the material reaches a precipitation temperature range from 400 to 700 °C.

During the study of TWIP steels for application in the mining industry, the authors had difficulties understanding the presence of carbides during heat treatment processes. Looking at the scientific literature, only a few articles related to manganese carbide precipitation were found. Due to this difficulty, the authors specifically oriented their research to obtaining an understanding of the precipitation kinetics of carbides in high-Mn TWIP steels.

Based on the previous discussion, Fe-Mn-carbides in a typical TWIP steel (Fe-22Mn-0.45C) with three cold-deformation levels (40, 60 and 80%), subjected to various annealing temperatures for different times, were characterized in this work. Their chemical features, morphology, precipitation sites, quantity, and size were also included. In addition, precipitation kinetic curves were determined as a function of room-temperature pre-deformation, obtaining the temperature–time–precipitation (TTP) curves for 0.1% (vol.) of carbides. Finally, austenitic grain size and microhardness were also measured.

## 2. Materials and Methods

Steel sample preparation: the steel was melted, alloyed and cast at the Metallurgical Department of the University of Santiago. The 25 kg ingot was forged at 1200 °C and homogenized for two hours in an electric furnace at Forjados Chile S.A., followed by hot rolling at a thickness of from 22 mm to 7.2 mm. After hot rolling, the steel was quenched in a salt bath at 350 °C, ensuring the absence of Fe-Mn-Carbides since, according to the literature [22], they only precipitate between 400 and 700 °C. To achieve different deformation levels, the steel was cold rolled to 40%, 60%, and 80% reductions in thickness, reaching 4.32 mm, 2.88 mm and 1.44 mm thicknesses. Samples were cut to 10 mm × 10 mm sizes for annealing at temperatures between 500 °C and 650 °C, with intervals of 25 °C, for 0.5, 1, 2, 4, 8 and 24 h. In this way, 108 conditions were analyzed. Surface decarburization was prevented by heating the samples in a box containing high-carbon steel shavings.

The chemical composition of the steel was determined in a Spectro MAXx optical emission Spectrometer according to ASTM E415 [24].

The austenite was characterized with an Olympus optical microscope. Samples were prepared with conventional mechanical polishing and etching by immersion in 3% Nital for 6 s. Curves of measured austenite grain size were plotted corresponding to 24-h annealing against annealing temperature for the three cold-deformation levels.

Scanning Electron Microscopy (SEM, JEOL JSM-6010 LA, Tokyo, Japan) was used to evaluate the presence and morphology of Fe-Mn-carbides. The carbides’ chemical composition was evaluated using Energy-dispersive X-ray spectroscopy (EDS, JEOL JED-2300, Tokyo, Japan) in SEM. In addition, the backscattered electron composition technique (BEC) was used for a quantitative evaluation of carbides. Three images were taken in the rolling direction on each sample, covering an area of 5000 μm^2^. The images were digitally treated with Image-Pro-Plus-6 software [25] to clearly highlight carbides from the matrix.

The amount of (Mn,Fe)_3_C in equilibrium for the analyzed temperature range was estimated using the software Thermocalc with the TCFE7 database.

Nine microhardness measurements were created according to ASTM E384 [26] on each sample, using a Struers Microhardness Duramin 1 tester with a force of 2.94 N.

The kinetics curve of Mn_3_C precipitation was modelled using a classical Johnson–Mehl–Avrami–Kolmogorov (JMAK) algorithm, as described by Kohout [27]:(1)$$X=1-exp\left(-Bt^{k}\right)$$
(2)$$B=B_{0}exp\left(-\frac{Q}{RT}\right)$$
where X represents the Mn_3_C precipitated fraction at temperature T and time *t*, $B_{0}$ and k are material constants. X was obtained by means of quantitative optical and/or electronic microscopy.

Curve temperature vs. time to reach 0.1% (vol.) of Fe-Mn-carbides precipitates was also determined.

## 3. Results

Figure 1 shows the amount of equilibrium (Mn,Fe)_3_C in the matrix as a function of temperature for the TWIP steel of the composition shown in Table 1.

### 3.1. Metallographic Analysis

Figure 2a shows the microstructure of the forged and homogenized steel, revealing an undeformed equiaxial 100 μm of average-size austenitic grains. Figure 2b shows an image from the same sample, obtained by SEM-secondary electron image (SEI), where no Fe-Mn-carbides can be observed.

Figure 3 shows images of cold-rolled microstructures, where deformed zones are clearly observed for 40%, 60%, and 80% area reductions. Elongated grains are evident on the more deformed samples.

### 3.2. Identification of Fe-Mn-Carbides

Figure 4a shows particles encircled by a dark line due to chemical attack, which are mainly located at grain boundaries. The chemical composition of the particle shown in Figure 4a, obtained by EDS, shows 25 at% C, 30.75 at% Mn and 44.25 at% Fe, with the stoichiometry of a (Fe,Mn)_3_C (6.73 wt% C, 37.87 wt% Mn and 55.4 wt% Fe). The composition of this particle was the closest to the theoretical composition of many measurements. The EDS technique, as a semiquantitative method, only detects the approximate chemical composition. As an example, Figure 4c shows other particles with a similar composition; (63 at% Fe, 19 at% Mn, 18 at% C). Data contained in the diffractogram shown in Figure 4b confirmed that the particles corresponded to (Fe,Mn)_3_C. Consequently, all the observed particles with these characteristics were assumed to be carbides and included in the measured carbide amount. A minimum of 10 images were analyzed to ensure the statistical validity of the measurements.

Figure 5 shows a BEC image of a sample deformed 80% by cold-rolling and annealed for 24 h at 575 °C. The clear dots correspond to the Fe-Mn-Carbides and the dark grey areas correspond to the austenitic matrix of the TWIP steel.

### 3.3. Measurement of the Volumetric Fraction of Fe-Mn-Carbides

The measurement of the volumetric fraction of Fe-Mn-carbides on specimens heated between 400 and 700 °C was carried out using scanning electron microscope and computational image analysis (Image Pro Plus 6). Several tests were carried out to find the precipitation limits. Figure 6a shows a micrograph corresponding to a sample that was subjected to 80% cold-deformation and annealed at 500 °C for 72 h. Figure 6b shows a similar sample, annealed at 675 °C for 24 h.

The evolution of the volume fraction percent of Fe-Mn-carbide precipitates was analyzed as a function of cold-deformation, time and temperature for all conditions. Figure 7 shows the results for a sample that is 80% cold-deformed. Figure 8 shows the corresponding micrographs.

As shown in Figure 7, the presence of carbides was appreciable after annealing 40% cold-deformed samples for 4 h, and 60% cold-deformed samples for 2 h, while only 1 h was needed for the 80% cold-deformed samples.

The Johnson–Mehl–Avrami–Kolmogorov [27] equation parameters for the Fe-Mn-carbide precipitation were calculated from a linearization of the experimental results (Figure 9) and summarized in Table 2. Based on the obtained results, Figure 10 shows the temperature vs. time values needed to reach 0.1% (vol.) of Fe-Mn-carbides precipitates.

Fe-Mn-carbide precipitate size was correlated with cold-deformation and temperature for 24-h annealing, as shown in Figure 11. The average precipitate size after 24-h annealing at different temperatures was determined with Image-Pro-Plus-6 software [25].

Figure 12 shows the Vickers microhardness of austenite after annealing at different temperatures, deformation levels and annealing times. The initial microhardnesses for samples with 40%, 60% and 80% cold-deformation were 390, 475 and 550 HV, respectively.

The austenite grain size results for all cold-deformation values after 24 h annealing are shown in Table 3.

## 4. Discussion

Although Fe-Mn-carbides in TWIP steels are not frequently discussed in the literature, it is well known that they can precipitate during thermomechanical forming or welding process, weakening some mechanical characteristics, as described in the Introduction. The aim of this work was to contribute to the knowledge of their kinetics and thermodynamics, which is necessary information when carbide precipitation must be avoided.

As is well known, (Fe,Mn)_3_C heterogeneously precipitates on high-energy sites such as grain boundaries, free surfaces, interphase boundaries, stacking faults, dislocations, and vacancies [28]. Klinkenberg et al. (2004) [29] studied the precipitation of NbC in a HSLA steel and found that the NbC particles heterogeneously nucleate on dislocations, sub-grains, grains and phase boundaries. Therefore, cold-deformation can favor the precipitation of carbides because plastic deformation mechanisms introduce high-energy sites for their nucleation. On the other hand, De Las Cuevas et al. (2018) [30], working with a 60% cold-rolled TWIP steel of a chemical composition that is relatively close to the one used in this paper, found that (Fe,Mn)_3_C precipitated on austenite grain boundaries that are heterogeneous, which is consistent with the observations in this work.

According to the thermodynamical calculated equilibrium of a precipitated volume fraction of carbide, as shown in Figure 1, the amount of equilibrium carbides decreases with increasing temperature and becomes very low at temperatures over 650 °C. The precipitates’ volume fraction was also measured and modelled using Equations (1) and (2), obtaining the kinetic parameters for each: the activation energy Q, k-exponent and B_0_ coefficient. Datapoints for each annealing temperature can be approximated by two slopes, as Figure 9 shows for 80% cold-deformation. This behavior was already reported by Lothongkun et al. [31] for chromium carbide in Fe 30.8Ni 26.6Cr austenitic steels. They related the chromium carbides’ precipitation to the diffusion coefficients of Cr in the bulk and at the grain boundaries. The activation energies reported by the authors were 213.2 kJ/mol and 51.7 kJ/mol, respectively. In this work, the activation energy corresponding to the first stage (Table 2) was the highest, coinciding with the cited interpretation. The activation energy values corresponding to the first stage decreased when cold-deformation increased, and the lowest were close to the activation energy for the diffusion of Mn in γ iron, 276 kJ/mol [20,32]. Table 2 also shows that the second stage exhibited a lower activation energy and, according to the cited authors [31], was probably related to boundary grain diffusion. There are more defects in strongly deformed materials, so a larger diffusion is expected and, consequently, a lower activation energy, as Table 2 shows for both stages.

Figure 6a shows a micrograph corresponding to the absence of Fe-Mn-carbides in a sample that was submitted to 80% cold-deformation and annealing at 500 °C. Deformed grains, evidencing deformed twins (i) and partial recrystallization (ii), can only be observed in limited zones. Similar images were obtained in samples with 80% cold-deformation and annealing up to 500 °C. Figure 6b shows the absence of carbides on a similar sample that was annealed at 675 °C for 24 h. There was no evidence of precipitation at temperatures higher than 675 °C for the studied time range. These results established the studied temperature interval.

Figure 7 shows that the volume fraction of Fe-Mn-carbides grew for all annealing temperatures as cold-deformation and annealing times increased. The figure also shows a lower volume percentage at higher temperatures for 80% cold-deformation, which is consistent with the fact that undercooling is increased at lower temperatures, despite the Mn diffusion being faster at higher temperatures, and also because they are close to the equilibrium amount. The interaction between nucleation, growth, and equilibrium can explain this behavior within a given temperature range. Table 2 summarizes the kinetic parameters for the three analyzed cases.

As shown in Figure 10, the precipitation of 0.1 vol% of Fe-Mn-carbides showed a typical “C” type appearance, in accordance with Kang [21]. The precipitation was much faster for 80% cold-deformed samples, and slower for smaller cold-deformations, displacing the curves to the right as cold-deformation decreased. At temperatures higher than that of the nose, that is, 575 °C for 80% cold-deformation and around 600 °C for lower cold-deformation values, the beginning of Fe-Mn carbides’ precipitation was delayed, as expected.

Fe-Mn-carbide precipitate sizes were correlated with the cold-deformation level and temperature used for 24 h annealing. Figure 11 shows that carbide sizes increased with increasing annealing temperatures and cold-deformation values, although, at 650 °C, Fe-Mn-carbide sizes abruptly decreased. The Fe-Mn-carbides size, which strongly depend on the annealing temperature, became stable after 24 h of annealing. The precipitate size for 80% cold pre-deformation more than doubled when the temperature increased from 525 °C to 625 °C, changing from 0.25 μm to 0.55 μm. When the annealing temperature was further raised to 650 °C, the precipitate size only reached around 0.25 μm and, as can be seen in Figure 7, the volume fraction was the smallest. This agrees with the fact that the equilibrium carbide amount at 650 °C is quite limited; see Figure 1. Carbide size was, for all temperatures, quite similar for 40% and 60% cold-deformation levels, although it was slightly larger for 80% cold-deformation.

The maximum presence of carbides, 2.5%vol, was observed on 80% cold-deformed samples and annealed at 525 °C for 24 h.

A complementary study of the recrystallization kinetics was performed and the main results are summarized in the following paragraphs. As a consequence of the high-strain-energy storage of deformation mechanisms, austenite grain size was approximately uniform and extremely fine for all samples, even for the highest annealing temperature and the lowest cold-deformation percentages (11.51 μm for 40% cold-deformation and 24-h annealing at 650 °C), as can be seen in Table 3.

During cold-deformation, an increase in hardness was observed in all samples. This increase in hardness is due to the increments in the dislocation density. These dislocations can interact and multiply using the Frank–Read mechanism. The sample with 80% cold-deformation had an initial hardness, after cold-deformation, that was higher than the samples with 60% and 40% cold-deformation, as previously reported.

Figure 12 shows that, with an increasing annealing time, microhardness decreases until a nearly constant value is reached, close to 220 Vickers, indicating that the matrix has completely recrystallized. The temperatures needed for a complete recrystallization after 24 h of annealing were 625 °C, 600 °C and 575 °C for 40%, 60% and 80% cold-deformation, respectively. At lower annealing temperatures, recrystallization was not complete after 24 h.

Figure 8 shows a temperature/time diagram of optical metallographies for temperatures between 525 and 625 °C and annealing times of 4, 8 and 24 h to aid in analysis of the evolution of precipitates and the presence of recrystallized grains. As annealing time increases, (Fe,Mn)_3_C carbides appear to have a banded structure due to Mn segregation. The maximum carbide quantity corresponds to 525 °C, decreasing as temperature increases for each annealing time, as can be seen in Figure 7.

Figure 8 also shows recrystallized grains in all the studied conditions; however, from hardness measurements, it can be deduced that recrystallization is only complete after annealing for 24 h at temperatures equal to or greater than 600 °C.

## 5. Conclusions

Manganese carbide precipitation kinetics were studied as a function of cold plastic deformation using a JMAK model. The kinetic model was divided into two stages. The activation energy corresponding to the first stage was always higher than the activation energy for the diffusion of Mn in γ iron. The second stage showed a lower activation energy than the first.

The precipitation of manganese carbides occurred in the temperature range between 525 and 650 °C. Carbide size increased with annealing temperatures and cold-deformation levels, although, at temperatures over 600 °C, Fe-Mn-carbide sizes abruptly decreased. Curves of temperature vs. time to 0.1 vol% Fe-Mn-carbides were built, showing that transformation is faster for higher cold-deformation levels. The maximum presence of carbides, 2.5%vol, was observed on 80% cold-deformed samples and annealed at 525 °C for 24 h. This amount was lower than that corresponding to the thermodynamic equilibrium.

## Figures and Tables

**Figure 1 materials-15-03748-f001:**
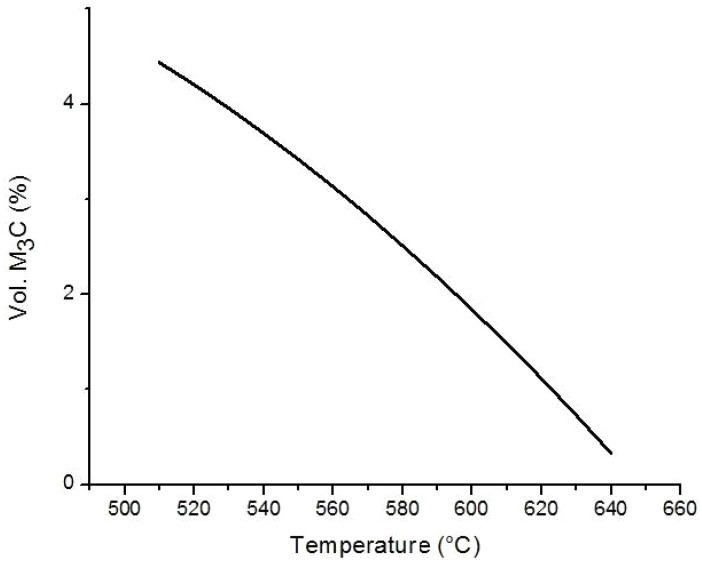
Thermodynamic equilibrium for precipitation of (Mn,Fe)_3_C (ThermoCalc).

**Figure 2 materials-15-03748-f002:**
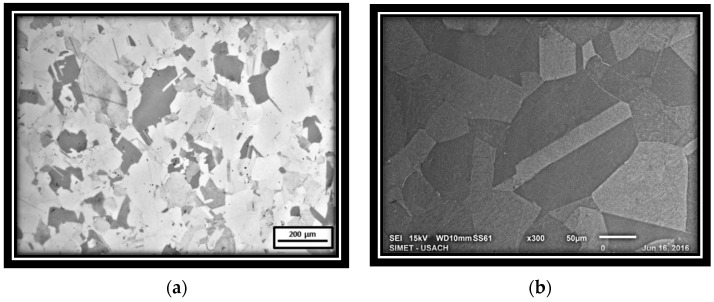
Microstructure of the studied steel: (**a**) OM image (**b**) SEM-SEI image.

**Figure 3 materials-15-03748-f003:**
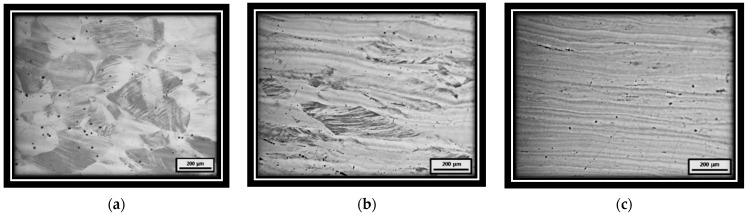
Microstructure of cold-deformed samples. (**a**) 40%; (**b**) 60%; (**c**) 80% reduction.

**Figure 4 materials-15-03748-f004:**
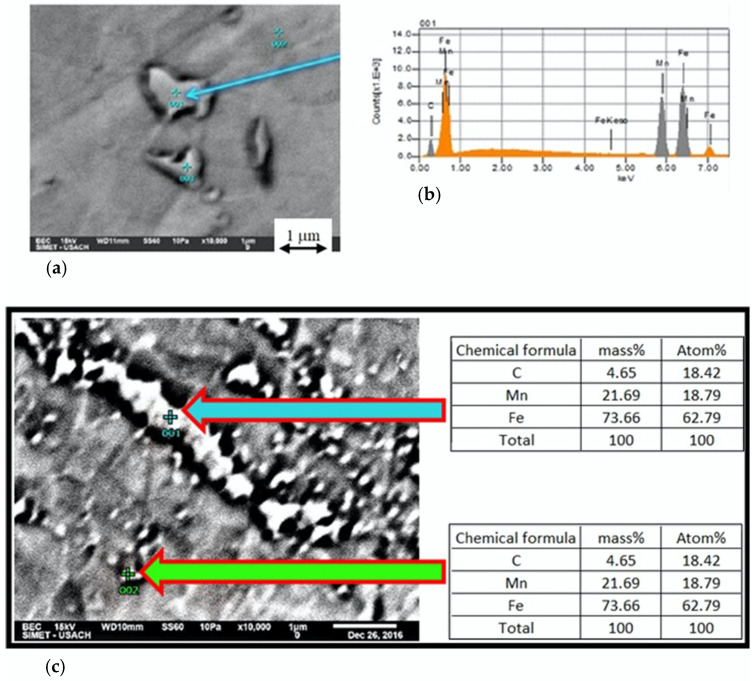
(**a**) Carbides, (**b**) diffractogram of a carbide, (**c**) carbides with EDS results.

**Figure 5 materials-15-03748-f005:**
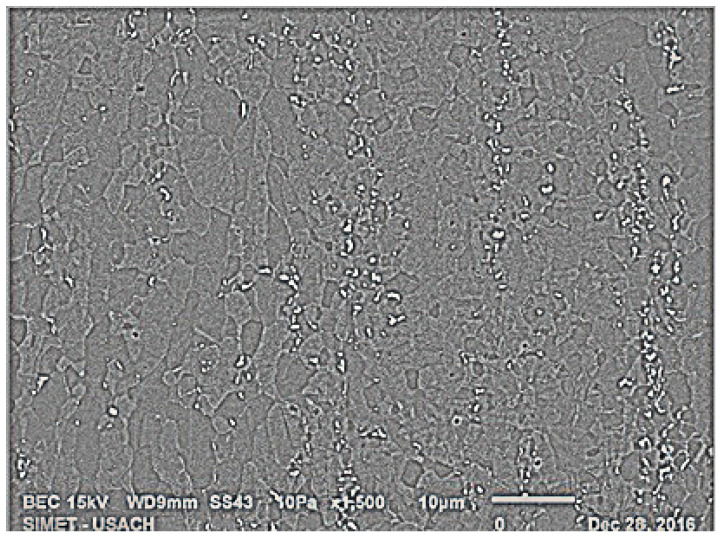
BEC image, showing Fe-Mn-Carbides in white and the austenitic matrix in grey. Cold-deformed 80% and annealed at 575 °C for 24 h.

**Figure 6 materials-15-03748-f006:**
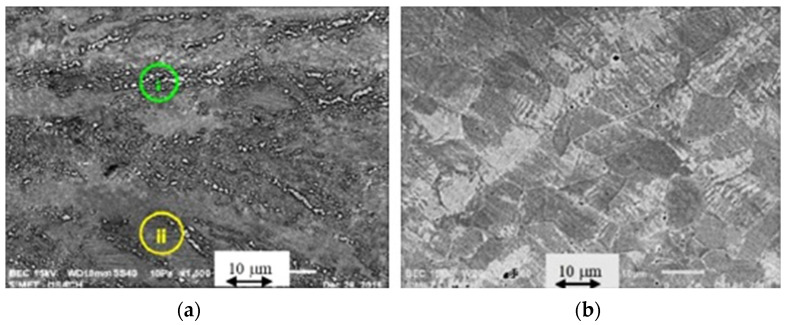
BEC images: (**a**) cold-deformed 80% and annealed at 500 °C for 72 h. (**b**) cold-deformed 80% and annealed at 675 °C for 24 h.

**Figure 7 materials-15-03748-f007:**
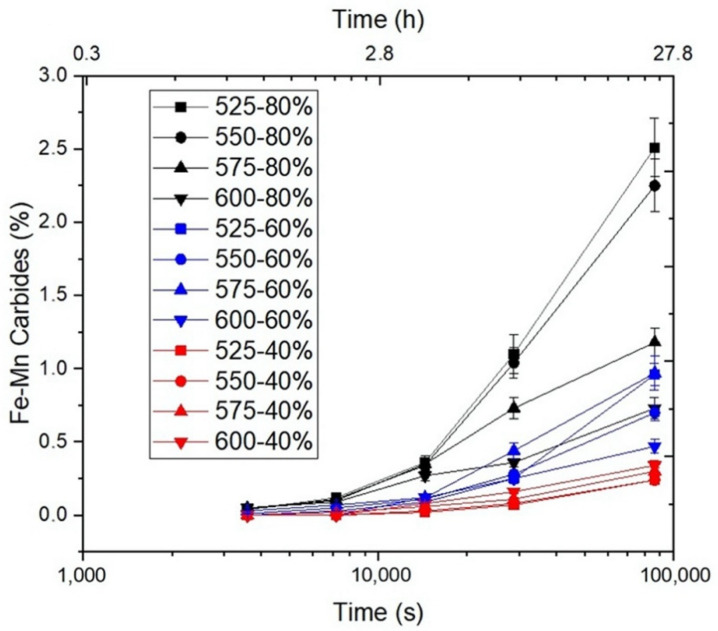
Volume percent of Fe-Mn-Carbide precipitation on 40, 60, and 80% cold-deformed samples annealed at different temperatures.

**Figure 8 materials-15-03748-f008:**
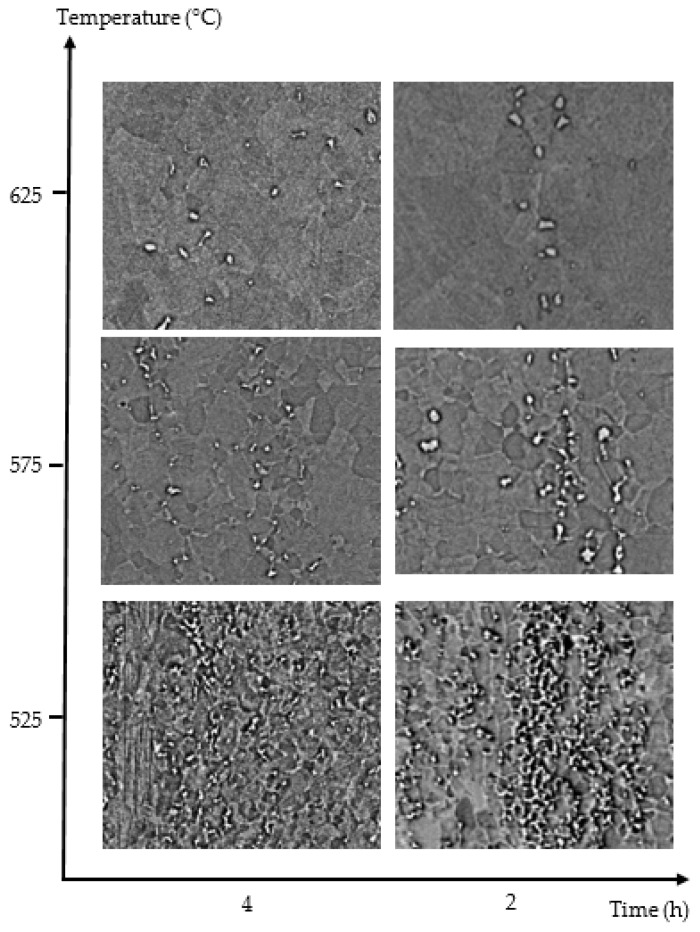
Microstructure evolution for 80% cold-rolled samples annealed at different temperatures and times.

**Figure 9 materials-15-03748-f009:**
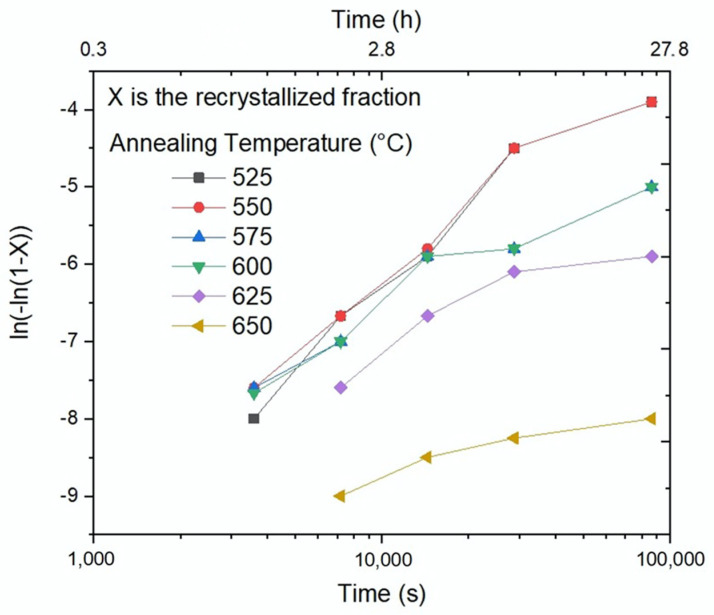
Linearization of the (Mn,Fe)_3_C volumetric fraction according to the kinematic JMAK model Equations (1) and (2). 80% cold-deformation.

**Figure 10 materials-15-03748-f010:**
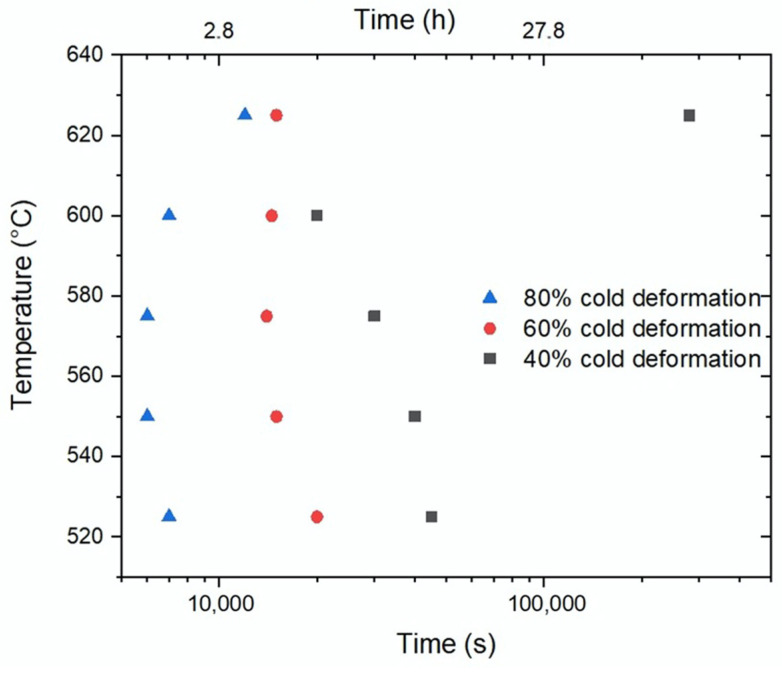
Temperature vs. time to 0.1 vol% (Fe,Mn)_3_C precipitation.

**Figure 11 materials-15-03748-f011:**
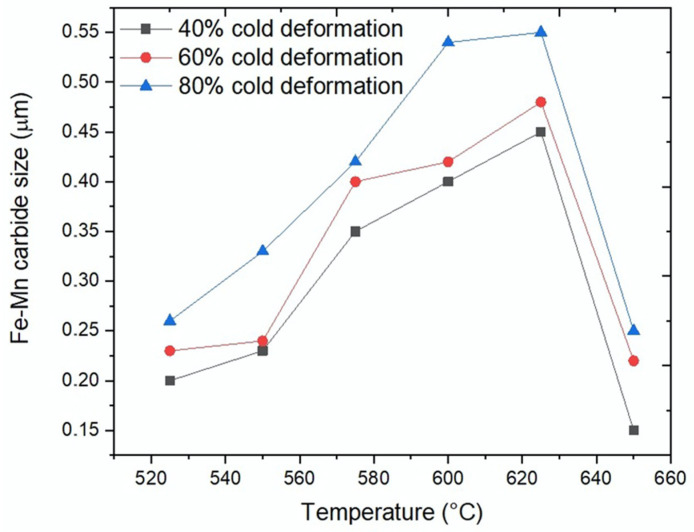
Evolution of (Fe,Mn)_3_C size with annealing temperature for different cold-deformation levels.

**Figure 12 materials-15-03748-f012:**
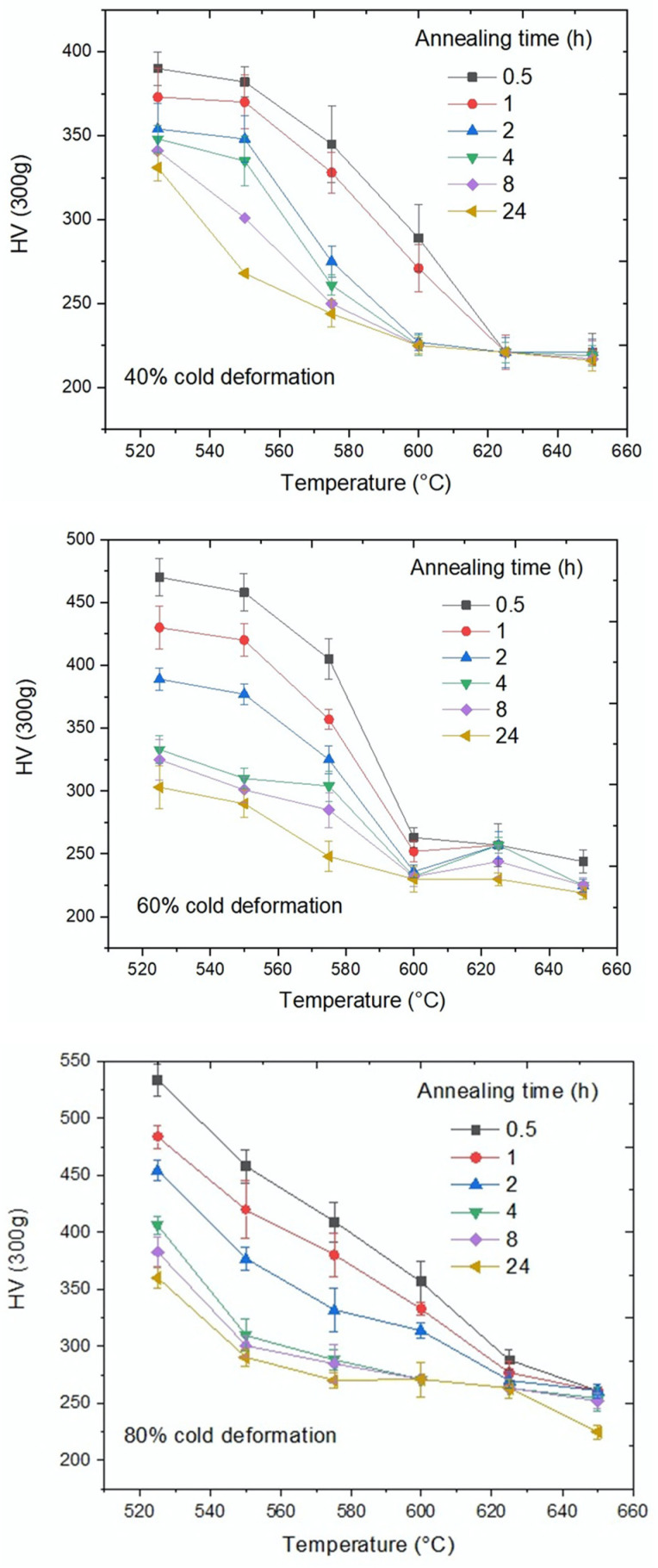
Vickers microhardness of Austenite vs. annealing temperature for different times.

**Table 1 materials-15-03748-t001:** Chemical composition of the TWIP steel (weight %).

%C	%Mn	%Si	%Cr	%P	%S	%Fe
0.45	21.94	0.21	0.16	0.008	0.002	balance

**Table 2 materials-15-03748-t002:** JMAK parameters for Fe-Mn carbide.

(a) Stage 1							
Temperature (°C)		525	550	575	600	625	650
40% cold-deformation	k	1.31	1.15	1.01	-	-	-
	Q	408.93 kJ/mol					
	B_0_	4.77 × 10^17^					
60% cold-deformation	k	1.46	1.34	1.25	-	-	-
	Q	326.38 kJ/mol					
	B_0_	1.48 × 10^12^					
80% cold-deformation	k	1.68	1.56	1.44	1.30	1.12	0.82
	Q	254.43 kJ/mol					
	B_0_	1.62 × 10^7^					
**(b) Stage 2**							
Temperature (°C)		525	550	575	600	625	650
40% cold-deformation	k	-	-	-	0.82	0.60	0.54
	Q	154.80 kJ/mol					
	B_0_	5.46 × 10^8^					
60% cold-deformation	k	-	0.84	0.83	0.73	0.67	0.47
	Q	104.14 kJ/mol					
	B_0_	2.12 × 10^00^ kJ/mol					
80% cold-deformation	k	0.76	0.75	0.63	0.56	0.45	0.22
	Q	78.85 kJ/mol					
	B_0_	6.73 × 10^−1^					

**Table 3 materials-15-03748-t003:** Austenite grain size (μm) for 40%, 60% and 80% cold-deformation after 24 h annealing at different temperatures.

Annealing Temperature (°C)	Austenite Grain Size (μm)		
	40% Deformation	60% Deformation	80% Deformation
525		1.76	0.87
550		1.98	1.30
575	3.68	2.65	2.48
600	4.00	3.78	3.52
625	7.52	6.49	5.27
650	11.51	9.24	7.17

## Data Availability

All data are included in this article.

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
