# Peer review of "Influence of Cold Deformation on Carbide Precipitation Kinetics in a Fe-22Mn-0.45C TWIP Steel"

_materials, 2022, doi:10.3390/ma15113748_

Round 1

Reviewer 1 Report

According to obtainef SEM images the size of carbides particle does not exceed 1 micron. At other hand the authors demonstrate the results of EDX  as compltely corresponding to chemical composition of M3C ( 25 at% C, 30.75 at% Mn and 44.25 at% Fe). It seems, it is necessary to indicate the condition of mesurements in order to reach such good results, especially for Carbon concentration. One of the main difficulties in such a study- to measure the size (and volume fracture) of quite small carbide particles. How the authors can be sure that most part of the particles was taken into account? also it is not indicated if the samples were etched before these measurements. Comparison between data presented on Fig. 7 and 9 demontrates some contradiction. If we take the highest value of carbide volume fraction (2.5 and 2.3 %) it gives the values of double logarithm visibly different, but no difference in two highest points on Fig. 9. Also, I do not undersand the text on Fig. 9 (X is recrystallized fraction).  The last question is about way to separate first and second stage according to table 2. 

Author Response

Comments of Reviewer 1

1.- The According to obtainef SEM images the size of carbides particle does not exceed 1 micron. At other hand the authors demonstrate the results of EDX  as compltely corresponding to chemical composition of M3C ( 25 at% C, 30.75 at% Mn and 44.25 at% Fe). It seems, it is necessary to indicate the condition of mesurements in order to reach such good results, especially for Carbon concentration.

Answer:

Results related to chemical composition of M3C obtained by EDS (or EDX) showed some dispersion. We did a lot of measurements and in the paper we presented the best of all this set of results. Certainly, the chemical composition had some dispersion, especially the carbon content. Despite this, we assumed that all particles having Mn, Fe and C corresponded to M3C. Finally, we included a note in the paper, clarifying this point and an example of other two carbide particles.

2.- One of the main difficulties in such a study- to measure the size (and volume fracture) of quite small carbide particles. How the authors can be sure that most part of the particles was taken into account? also it is not indicated if the samples were etched before these measurements.

Answer:

Effectively, it is not easy to measure the fraction of small particles. However, in order to measure the particle fraction with a statistical validity, a minimum of ten images were processed for each condition. A short note was inserted in the text of the paper with this information.

3.- Comparison between data presented on Fig. 7 and 9 demontrates some contradiction. If we take the highest value of carbide volume fraction (2.5 and 2.3 %) it gives the values of double logarithm visibly different, but no difference in two highest points on Fig. 9.

Answer:

  1. At 525ºC, for a volume fraction=2.5%, X=2.5/100=0.0025

the value of ln[-ln(1-X)]=ln[-ln(1-0.025)]= - 3.67

  1. At 550ºC, for a volume fraction=2.25%, X=2.3/100=0.0023

the value of ln[-ln(1-X)]=ln[-ln(1-0.0225)]= - 3,78

Figure 9 was replaced, including two significant figures after the comma.

4.- Also, I do not undersand the text on Fig. 9 (X is recrystallized fraction). 

Answer:

This was a mistake: X is the carbide fraction during recrystallization process. Figure 9 was replaced.

5.- The last question is about way to separate first and second stage according to table 2. ?   

Answer:

Curve ln[-ln (1-x)] shows two slopes. These slopes allow us to deduce the existence of two stages, in agreement with other authors such as Lothongkum et al. (ref 28).  The meaningful of each stage is discussed in lines 252 to 260. The activation energy for first stage is greater than that for second stage. The value of the activation energy for first stage is close to the activation energy for diffusion of Mn in g iron. This is consistent with the formation of Mn3C, that requires the diffusion of Mn in iron.

Yours faithfully,

On behalf of all the authors,

Reviewer 2 Report

This paper aims to study the influence of cold deformation in carbide precipitation kinetics in a Fe-22Mn-0.45C TWIP steel. The following comments are made: 

  1. It is recommended to show the microstructure and grain size of the cold-rolled samples, before and after annealing under a certain condition.
  2. The increase in the size or number of Fe-Mn-Carbide precipitates with increasing cold deformation percentage at a certain annealing condition should be indicated by the SEM.
  3. More evidence is needed to show matrix recrystallization under different annealing conditions. It is recommended to add some EBSD images. (Lines 301-314)
  4. The authors should discuss in detail the reason for increasing the hardness of the specimens annealed in similar conditions via increasing the cold rolling percentage before annealing (All aspects).

Author Response

This paper aims to study the influence of cold deformation in carbide precipitation kinetics in a Fe-22Mn-0.45C TWIP steel. The following comments are made: 

  1. It is recommended to show the microstructure and grain size of the cold-rolled samples, before and after annealing under a certain condition.

Answer:

Figure 3 shows the cold deformed samples at three cold-deformation conditions: 40%, 60% and 80%. Figure 6 shows the microstructure of the sample cold-deformed 80% and annealed at 500ºC and 675ºC for 24 h.

  1. The increase in the size or number of Fe-Mn-Carbide precipitates with increasing cold deformation percentage at a certain annealing condition should be indicated by the SEM.?  

Answer:

Yes, the increase in carbide sizes with increasing cold deformation can be determined by SEM. In this work, mainly optical microscopy was used in order to measure the size of precipitates, although in some cases, also SEM was used.

  1. More evidence is needed to show matrix recrystallization under different annealing conditions. It is recommended to add some EBSD images. (Lines 301-314)

Answer:

In a new stage of this research project, it is considered to carry out EBSD measurements. However, during the period of the investigation, EBSD was not available. The authors will develop this study during the present year.

  1. The authors should discuss in detail the reason for increasing the hardness of the specimens annealed in similar conditions via increasing the cold rolling percentage before annealing (All aspects).

Answer:

A brief note explaining the origin of the hardening in samples cold deformed was included in the text of the paper. Basically, the interactions between dislocations and the multiplication of dislocations, can explain this hardening process. After cold deformation and before annealing, no precipitation of carbides had occurred, so, the increment in hardness is only due to the increment in dislocation density.

Yours faithfully,

On behalf of all the authors,

Reviewer 3 Report

1/ Title is a little bit weird. Reconsider to change it.

2/ English must be improved troughtout the whole manuscript.

3/ Introduction and subsequent Discussion are focused only on high-C high-manganese steels. Please note that there are also low-C high-manganese steel, in which a numer of carbides is significantly smaller. This information should be provided for clarity, what is a role of carbon content in high-Mn steels. The alloys are also alloyed with Al and microadditions: Ti, Nb, V, which also form carbides

4/ Enhance a novelty of the study because the topi cis already known in literature

5/ Thermocalc – provide the database and other important information on Software

6/ What are these color grains in Figure 2a and Figure 3 ?

7/ Since particles are the mots important part of the manuscript theie identification is of primary importance. The images in Figure 4 are of very low quality. Moreover, more exaples from different temperatures should be provided.

8/ Figure 7. The method is very risky based only on optical image analyis. The errors bars and statistic data must be completed.

9/ A low quality of Figure 8 is unacceptable. It must be done in a much better way

10/ I can not find results of X-ray analysis, which was written in Methodes section. This must be provided.

11/ Conclusions must be rewritten as one or two concise paragraphs instead of numerous separated sentences

Author Response

All the suggestions were considered.

Round 2

Reviewer 2 Report

The manuscript can e published in its current form. 

Author Response

Reviewer 2:

From reviewer 2:

"The manuscript can e published in its current form."

Thank you very much for your comment.

On behalf the authors

Alberto Monsalve G.

Reviewer 3 Report

The authors improved most of the comments indicated by me. Some parts still need the improvement. For example, they added some information on low-C high-Mn steels, however, they did not support it by literature, which can be easily found in open literature. Moreover, the quality of micrographs should be improved further as much as possible.

Author Response

Reviewer 3:

We  leave the figures as best as possible.

Thank you for your comments.

On behalf on the authors

Alberto Monsalve
